# Enhancing Colorectal Cancer Immunotherapy: The Pivotal Role of Ferroptosis in Modulating the Tumor Microenvironment

**DOI:** 10.3390/ijms25179141

**Published:** 2024-08-23

**Authors:** Yanqing Li, Xiaofei Cheng

**Affiliations:** 1Department of Pathology, The Second Affiliated Hospital, Zhejiang University School of Medicine, Hangzhou 310009, China; yanqingli@zju.edu.cn; 2Department of Colorectal Surgery, The First Affiliated Hospital, Zhejiang University School of Medicine, Hangzhou 310003, China

**Keywords:** ferroptosis, immunotherapy, colorectal Cancer (CRC), tumor microenvironment (TME), ferroptosis-related genes (OFRGs)

## Abstract

Colorectal cancer (CRC) represents a significant challenge in oncology, with increasing incidence and mortality rates worldwide, particularly among younger adults. Despite advancements in treatment modalities, the urgent need for more effective therapies persists. Immunotherapy has emerged as a beacon of hope, offering the potential for improved outcomes and quality of life. This review delves into the critical interplay between ferroptosis, an iron-dependent form of regulated cell death, and immunotherapy within the CRC context. Ferroptosis’s influence extends beyond tumor cell fate, reshaping the tumor microenvironment (TME) to enhance immunotherapy’s efficacy. Investigations into Ferroptosis-related Genes (OFRGs) reveal their pivotal role in modulating immune cell infiltration and TME composition, closely correlating with tumor responsiveness to immunotherapy. The integration of ferroptosis inducers with immunotherapeutic strategies, particularly through novel approaches like ferrotherapy and targeted co-delivery systems, showcases promising avenues for augmenting treatment efficacy. Furthermore, the expression patterns of OFRGs offer novel prognostic tools, potentially guiding personalized and precision therapy in CRC. This review underscores the emerging paradigm of leveraging ferroptosis to bolster immunotherapy’s impact, highlighting the need for further research to translate these insights into clinical advancements. Through a deeper understanding of the ferroptosis-immunotherapy nexus, new therapeutic strategies can be developed, promising enhanced efficacy and broader applicability in CRC treatment, ultimately improving patient outcomes and quality of life in the face of this formidable disease.

## 1. Introduction

Colorectal cancer (CRC) ranks among the top three in both incidence and mortality rates globally [1]. Most CRCs originate from polyps, beginning with abnormalities in glandular crypts and progressing into polyps and then cancer. This process can take 5 to 15 years, with villous adenomas exhibiting a high malignancy rate of up to 40% [2]. In recent years, there has been a significant increase in the incidence and mortality rates of CRC among young adults aged 18 to 49 years across nearly all regions worldwide, signaling an alarming trend that underscores the urgent need for effective treatment modalities [3].

Sex differences in the progression of CRC have become an increasingly important area of research. Studies show that men and women exhibit significant differences in CRC incidence, progression, and response to treatment. Generally, men have a higher incidence of CRC, which may be linked to lifestyle-related risk factors such as smoking, alcohol consumption, and diet [4]. In contrast, women may benefit from the protective effects of hormones like estrogen, which may delay CRC onset. However, this protective effect diminishes after menopause, leading to a convergence in CRC risk between men and women.

At the genetic level, there are sex-based differences in the frequency and types of mutations in CRC-related genes, such as *APC* and *TP53*, which can influence tumor behavior and treatment response [5]. Estrogen, for example, is thought to exert anti-tumor effects by interacting with intracellular signaling pathways that inhibit CRC development [6]. Additionally, estrogen may modulate the immune microenvironment, potentially enhancing the effectiveness of immunotherapy [7]. Sex differences also extend to the immune response, with women generally exhibiting a stronger immune reaction compared to men [8]. This heightened immune response could lead to better outcomes in immunotherapy for women but also increases the risk of immune-related side effects.

Immunotherapy has shown promising prospects in cancer treatment, with the potential to enhance treatment outcomes, prolong survival, and improve patients’ quality of life. The aspiration for long-term survival with the disease or complete tumor eradication is becoming an increasingly tangible goal, thanks to immunotherapy. The significance of immunotherapy in cancer treatment is evident in several aspects. Firstly, it is highly specific and efficient, mobilizing the body’s immune cells to recognize and combat cancer, thereby indirectly eliminating and controlling the disease with minimal side effects, ensuring safety and efficacy [9]. Secondly, immunotherapy encompasses a diverse range of approaches, such as molecular targeted therapy, immune checkpoint inhibitors (ICIs), adoptive cell therapy, cytokine therapy, and cancer vaccines, among others [10]. Unlike traditional treatments such as surgery, radiotherapy, and chemotherapy, immunotherapy does not directly kill cancer cells. Instead, it activates the body’s immune cells, enhancing the immune system’s ability to combat the disease. With advances in tumor immunology research and continuous technological advancements, immunotherapy is poised to achieve new breakthroughs in tumor eradication, aspiring to become a mainstream method in cancer treatment [11].

When discussing CRC and immunotherapy, understanding the concept of ferroptosis becomes particularly relevant. Ferroptosis, an iron-dependent form of regulated cell death, has garnered significant interest in recent years due to its potential role in tumor immunity. There exists a complex interplay between ferroptosis and tumor immunity. On the one hand, ferroptosis can influence the immunogenicity of tumor cells, thereby affecting the recognition and attack by immune cells [12]. On the other hand, ferroptosis also impacts the activity of immune cells, as iron plays a crucial role in their activation, proliferation, and functionality [13]. Moreover, the relationship between ferroptosis and ICIs, a class of drugs used to enhance the immune system’s attack on tumors, has been elucidated. Research suggests that ferroptosis may be associated with the efficacy of ICIs. Ferroptosis can affect the expression of certain molecules on tumor cells, thereby influencing the expression of immune checkpoints and subsequently impacting the effectiveness of immunotherapy [14]. Furthermore, the metabolic pathways of immune cells are crucial for their functionality. Ferroptosis can influence the metabolic pathways of immune cells, affecting their activity, proliferation, and functionality within the TME [15]. Consequently, ferroptosis may serve as a critical factor in regulating the role of immune cells in the TME.

In conclusion, the concept of ferroptosis holds promise in reshaping our understanding of tumor immunity and immunotherapy. Its intricate interactions with tumor cells and immune cells underscore its potential as a novel therapeutic target for enhancing the efficacy of immunotherapy in CRC and other malignancies.

## 2. Ferroptosis: Understanding Its Mechanism and Impact

Introduced by Brent R. Stockwell from Columbia University in 2012, ferroptosis is recognized as a unique, regulated cellular demise mechanism driven by iron dependency, setting it apart from apoptosis, necrosis, and autophagy [16]. It pivots around the essential roles of glutathione depletion and the inactivation of glutathione peroxidase 4 (*GPX4*), hindering the cellular capacity to neutralize lipid peroxides. This deficiency fosters an environment ripe for iron accumulation, escalating lipid oxidation, and the formation of reactive oxygen species (ROS), culminating in cell death [17]. Distinguished by iron buildup, lipid peroxidation, and enhanced ROS levels, ferroptotic cells also display genetic shifts in iron homeostasis and lipid peroxidation regulation. Microscopically, these cells are marked by diminished mitochondrial volume, with notable alterations in membrane and cristae structures, while nuclear morphology remains largely unaffected [18].

Ferroptosis intersects with several metabolic pathways, including those involving amino acids, iron, and polyunsaturated fatty acids, along with the synthesis of glutathione, phospholipids, NADPH, and coenzyme Q10. Its implication in various mammalian degenerative conditions and diseases highlights its pathological significance [19].

Central to triggering ferroptosis is the rampant accumulation of lipid peroxides, propagated through two primary routes: enzymatic lipid peroxidation and the iron-catalyzed Fenton reaction. The enzymatic pathway transforms polyunsaturated fatty acids (*PUFAs*) into reactive lipid peroxides, with arachidonic (AA) and adrenic acids as key PUFA contributors [20]. This process is meticulously orchestrated by enzymes such as acyl-CoA synthetase long-chain family member 4 (*ACSL4*), lysophosphatidylcholine acyltransferase 3 (*LPCAT3*), and lipoxygenases (*LOXs*), which sequentially activate and oxidize AA to form lipid peroxides, thus playing pivotal roles in ferroptosis induction [21]. Conversely, the Fenton reaction, facilitated by free iron ions, produces lipid peroxides from peroxyl radical interactions with lipids. This mechanism is exacerbated by fluctuations in cellular iron levels, intensifying lipid peroxide formation and propelling the cell toward ferroptosis [22]. To counteract lipid peroxide accumulation, cells employ *GPX4*, which utilizes glutathione (*GSH*), synthesized from imported cysteine, to revert lipid peroxides to their benign state. Given *GPX4′*s exclusivity in this reduction, its role is deemed indispensable in the ferroptotic pathway, making it a prime target for therapeutic interventions [23].

Beyond the *GSH* pathway, the coenzyme Q10 (*CoQ10*)-dependent reduction pathway also mitigates ferroptosis. Here, the ferroptosis suppressor protein FSP1 plays a crucial role, leveraging NADH to restore *CoQ10* to its antioxidative ubiquinol form, thus reducing lipid peroxides [24]. Other noteworthy mechanisms include the *GCH1* pathway, producing the antioxidant BH4 and facilitating *CoQ10* synthesis, and *DHODH’s* contribution, paralleling FSP1′s function in the mitochondria to dampen lipid peroxide production, emphasizing the complexity and multifaceted nature of cellular defenses against ferroptosis [25]. Figure 1 illustrates the primary signaling pathways regulating ferroptosis.

## 3. Effect of Tumor Microenvironment (TME) on Ferroptosis

Hypoxia, a hallmark of the TME, induces changes in iron metabolism that impact ferroptosis regulation. Under hypoxic conditions, iron regulatory protein 2 (*IRP2*) is upregulated post-translationally, leading to increased expression of iron transporters (TFRC, *SLC11A2*) and decreased expression of iron storage proteins (FTH) [26]. Carbonic anhydrase IX (CAIX), a key regulator of tumor hypoxia, controls intracellular pH and prevents cancer cells from undergoing ferroptosis by modulating the cystine/glutamate antiporter xCT [27,28]. Hypoxia-induced factors such as *HIF-1α/lncRNA-PMAN* and the *CBSLR/CBS* signal axis further protect cancer cells from ferroptosis by regulating key genes involved in iron metabolism [29]. Additionally, hypoxia reduces the expression of NCOA4 in macrophages, leading to increased ferritin levels and decreased susceptibility to ferroptosis [22,30]. Recently, dual hypoxia-sensitive polymeric nanocarriers have been designed to sensitize hypoxic tumor cells to ferroptosis by depleting NADPH, *GSH*, and Trx, thereby inhibiting *GPX4* activity and enhancing the efficacy of ferroptosis-inducing agents, specifically in hypoxic tumors [31,32].

Lactate accumulation in the TME, resulting from cancer cell glycolysis, has been found to inhibit ferroptosis. High levels of lactic acid upregulate the expression of hydroxycarboxylic acid receptor 1 (*HCAR1*) and monocarboxylate transporter 1 *(MCT1*), thereby reducing lipid peroxidation and inhibiting ferroptosis in hepatoma cells [33]. Targeting the HCAR1/MCT1 axis enhances ferroptosis sensitivity, suggesting a potential therapeutic strategy for overcoming lactate-mediated ferroptosis resistance in cancer treatment [33].

Furthermore, cancer-related inflammation within the TME can either promote or inhibit tumor ferroptosis. Proinflammatory factors such as *IL-6* and *HMGB1* have been implicated in regulating ferroptosis in various cancer types through different signaling pathways, including the *JAK2/STAT3* pathway and the *RAS-JNK/p38* pathway [34]. Additionally, non-coding RNAs like miR-539 and circRNAs like *CircABCB10* and *Circ-IL-4* receptors have been shown to modulate ferroptosis by targeting key regulators such as xCT and *GPX4*, thereby influencing cancer progression [35].

Recent studies have shed light on the intricate relationship between ferroptosis and immune effector cells within the TME, underscoring its implications for cancer immunotherapy. CD8+ T cells, pivotal players in antitumor immunity, secrete cytokines such as *IL-2*, *IL-12*, and *IFNγ* within the TME, enhancing their ability to target and eliminate tumor cells. Interestingly, *IFNγ* produced by activated CD8+ T cells inhibits the expression of *SLC3A2* and Solute Carrier Family 7 Member 11 *(SLC7A11)*, components of the cystine/glutamate antiporter system xc-, promoting lipid peroxidation and ferroptosis in tumor cells [36,37]. Additionally, *IFNγ*, in conjunction with arachidonic acid, induces tumor cell ferroptosis through *ACSL4*, highlighting the potential of targeting tumor ferroptosis metabolism to enhance cancer immunotherapy [38,39].

Moreover, Oxidized Low-Density Lipoprotein (*oxLDL*) in the TME induces ferroptosis and p38 phosphorylation in CD8+ T cells via CD36-dependent mechanisms. Activation of p38 leads to CD8+ T cell death, suppression of *IFNγ* and TNFα production, and depletion of CD8+ T cells, impairing antitumor immunity. Notably, combination therapy involving anti-PD-1 antibodies and CD36 deletion in CD8+ T cells exhibits superior antitumor effects, suggesting that targeting CD36 and ferroptosis could enhance T cell-based immunotherapy efficacy [40,41,42]. NK cells, crucial for tumor surveillance and immunotherapy, exert cytotoxic effects on tumor cells through perforin, granzyme, and *IFNγ* release. Furthermore, activating mitochondrial apoptosis in cancer cells enhances NK cell-mediated killing, and the synergistic effect of BH3 mimetics and ferroptosis induction enhances cancer cell death. Clinical-grade iron oxide nanoparticles induce ferroptosis in prostate cancer cells, activating NK cells and augmenting their cytotoxic function. Combining ferroptosis induction with NK cell therapy results in tumor regression, indicating that ferroptosis enhances NK cell activity [42,43].

However, while ferroptosis releases cytokines and damage-associated molecular patterns (DAMPs), it does not activate antitumor immune responses. Moreover, ferroptosis inhibits the cross-presentation of soluble antigens to dendritic cells (DCs), impairs DC maturation, and inhibits DC-mediated phagocytosis of tumor cells. Consequently, cancer cell ferroptosis may not represent an immunogenic form of cell death, highlighting the complex interplay between ferroptosis and immune responses within the TME [44,45].

To fully understand the complex regulatory networks of ferroptosis in the TME, it is essential to integrate interdisciplinary approaches, including computational biology, bioinformatics, and systems biology. These fields offer powerful tools and methodologies to dissect the intricate interactions and pathways involved in ferroptosis, providing deeper insights into its role in cancer therapy.

Computational biology utilizes mathematical models and computational techniques to simulate biological processes. In the context of ferroptosis, computational models can predict the dynamics of lipid peroxidation, iron metabolism, and antioxidant defenses under various conditions. By simulating these processes, researchers can identify key regulatory nodes and potential therapeutic targets. For instance, computational models can help predict how different ferroptosis inducers interact with cellular pathways, aiding in the design of more effective and specific treatments [46].

Bioinformatics involves the application of computational tools to analyze biological data. High-throughput omics technologies, such as genomics, transcriptomics, proteomics, and metabolomics, generate vast amounts of data that can be analyzed to uncover the molecular mechanisms underlying ferroptosis. Bioinformatics approaches can identify gene expression patterns, protein-protein interactions, and metabolic alterations associated with ferroptosis [47]. For example, integrating transcriptomic data from CRC patients with ferroptosis-related gene expression profiles can reveal biomarkers for patient stratification and treatment response prediction [48,49].

Systems biology adopts a holistic approach to understanding biological systems by integrating data from multiple sources to build comprehensive models. In ferroptosis research, systems biology can elucidate the complex interactions between different cellular components and pathways within the TME. This approach can identify emergent properties and network behaviors that are not apparent from studying individual components in isolation [50]. Systems biology models can also simulate how interventions, such as ferroptosis inducers or inhibitors, impact the overall system, providing insights into potential therapeutic strategies and their systemic effects [51].

Combining computational biology, bioinformatics, and systems biology can lead to a more comprehensive understanding of ferroptosis in the TME. For instance, computational models can be informed by bioinformatics analyses of omics data, while systems biology can integrate these models to simulate complex interactions within the TME [52]. This interdisciplinary approach can identify novel regulatory mechanisms, potential biomarkers, and therapeutic targets, ultimately enhancing the efficacy of ferroptosis-based therapies in CRC [53].

In CRC research, predictive modeling can predict patient-specific responses to ferroptosis-based therapies, allowing for personalized treatment plans. Bioinformatics analyses can identify biomarkers associated with ferroptosis sensitivity and resistance, facilitating patient stratification and monitoring [50]. Systems biology can reveal key regulatory nodes and pathways that can be targeted to enhance ferroptosis and improve therapeutic outcomes. By leveraging these interdisciplinary approaches, researchers can gain a deeper understanding of the regulatory networks of ferroptosis in the TME, paving the way for more effective and targeted cancer therapies.

In conclusion, the relationship between ferroptosis and the TME is intricate and multifaceted, impacting not only the survival and death of tumor cells but also the function of immune cells, thereby influencing tumor development and treatment response. Table 1 summarizes the regulatory interactions between ferroptosis and the TME, highlighting the mechanisms and effects. Understanding these interactions is crucial for developing novel cancer therapeutic strategies.

## 4. The Role of Ferroptosis in Immunotherapy for Cancer

The role of ferroptosis, a non-apoptotic form of cell death induced by iron-dependent lipid peroxidation, is increasingly recognized in cancer immunotherapy. Not only does ferroptosis directly cause tumor cell death, but it also enhances the efficacy of immunotherapy by modulating the activity of immune cells within the TME.

In the treatment of melanoma, activation of ferroptosis sensitizes tumor cells to chemotherapy and enhances immunotherapy efficacy by regulating the activity of immune cells within the TME. For instance, ferroptosis-induced tumor cell death promotes the polarization of M1-like macrophages while inhibiting the functions of M2-like macrophages that promote tumor growth and metastasis [54,55]. In pancreatic cancer therapy, ferroptosis induction selectively kills tumor cells and activates the immune system, bolstering antitumor immune responses [56,57,58]. Studies have shown that ferroptosis inducers such as RSL3 and FIN56 increase tumor cell sensitivity to radiotherapy, thereby improving treatment efficacy when used in combination [59,60]. Furthermore, ferroptosis promotes the release of DAMPs from tumor cells, activating dendritic cells and T cells to enhance tumor-specific immune responses [61,62]. In breast cancer therapy, ferroptosis induction also shows promise. When combined with ICIs, it effectively enhances tumor cell ferroptosis and improves treatment outcomes. For example, CD8+ T cells suppress the expression of *SLC7A11* and *SLC3A2* in tumor cells via IFNγ release, increasing tumor cell sensitivity to ferroptosis [37,63]. Moreover, ferroptosis inducers such as sulfasalazine and elastin enhance breast cancer cell sensitivity to chemotherapy drugs [60].

In summary, the role of ferroptosis in cancer immunotherapy is multifaceted. It not only directly induces tumor cell death but also enhances immunotherapy efficacy by modulating immune responses within the TME. Induction and regulation of ferroptosis hold promise for improving treatment outcomes in various types of cancer, including melanoma, pancreatic cancer, breast cancer, and CRC.

## 5. The Challenges of Immunotherapy in CRC

Immunotherapy holds immense promise in the treatment of CRC, yet it faces several significant challenges. Firstly, its applicability is limited to a subset of patients, primarily those with microsatellite instability-high (MSI-H) or deficient mismatch repair (dMMR) status. This restriction means that a large proportion of CRC patients, particularly those with microsatellite stable (MSS) or proficient mismatch repair (pMMR) status, currently do not benefit from immunotherapy [64]. Expanding the eligible patient population for immunotherapy remains a major hurdle [65,66]. Secondly, accurately predicting treatment efficacy poses a challenge. MSI-H/dMMR status is currently the best predictor of immunotherapy response. However, even within this subgroup, only about 40% of patients respond effectively, leaving a significant portion without benefit. Improving the accuracy of predicting which patients will benefit from immunotherapy is another critical challenge [65,67,68]. Thirdly, treatment strategies require optimization. While combination immunotherapy with *CTLA-4* inhibitors has shown good efficacy in MSI-H/dMMR patients, it also comes with increased toxicity. Balancing efficacy and reducing side effects through optimized treatment strategies is a current imperative [65,69,70]. Fourthly, transforming “cold tumors” like MSS CRC into “hot tumors” that respond well to immunotherapy presents a substantial challenge. Efforts to achieve this transformation have not yet succeeded. Effectively converting “cold tumors” into “hot tumors” represents another significant challenge [71,72,73,74]. Lastly, there is a need for the development of novel immunotherapy strategies and drugs. While some immunotherapy agents like PD-1 inhibitors and CAR-T therapy are currently used in CRC treatment, they have limitations [75,76]. Developing new and more effective immunotherapy strategies and drugs to meet the needs of a broader patient population is an ongoing and future imperative.

Addressing these challenges is crucial for advancing the field of immunotherapy in CRC treatment. Interestingly, emerging research suggests that ferroptosis may offer a potential avenue for addressing some of these challenges.

## 6. The Role of Ferroptosis in Immunotherapy for CRC

In CRC, ferroptosis not only influences the fate of tumor cells but also exerts profound effects on the efficacy of immunotherapy by altering the composition and function of the TME. Studies investigating the expression patterns of OFRGs in CRC have revealed the crucial role of ferroptosis in regulating immune cell infiltration and the formation of the TME. These studies have found a close correlation between the expression levels of OFRGs and the sensitivity of tumors to immunotherapy. In immune-desert tumors characterized by low immune cell infiltration, induction of ferroptosis can increase immune cell infiltration, particularly CD8+ T cells, thereby enhancing tumor immunogenicity [77]. Conversely, in immune-inflamed and immune-excluded tumors, ferroptosis may influence the efficacy of immunotherapy by modulating the number and function of immune suppressor cells such as regulatory T cells (Tregs) and myeloid-derived suppressor cells (MDSCs) [78,79]. Thus, ferroptosis not only acts directly on tumor cells but also indirectly enhances the efficacy of immunotherapy by modulating immune cell activity and the state of the TME [80,81]. Furthermore, the roles of molecules such as Coactivator Associated Arginine Methyltransferase 1 (*CARM1*)and heat shock protein family B (small) member 1 *(HSPB1)* in regulating ferroptosis further emphasize the importance of ferroptosis in the tumor immune microenvironment. CARM1 promotes tumor progression by methylating *ACSL4* while inhibiting CARM1 can increase tumor cell sensitivity to ferroptosis, thereby promoting immunogenic cell death of tumor cells [82]. On the other hand, HSPB1 negatively regulates ferroptosis by reducing iron-mediated ROS production [83]. These findings suggest that targeting these molecules can modulate tumor cell sensitivity to ferroptosis, thereby enhancing the effectiveness of immunotherapy [84,85].

The combination of ferroptosis inducers and immunotherapy represents a promising approach to cancer treatment, particularly in CRC therapy. Ferrotherapy is an emerging cancer treatment modality that induces ferroptosis by catalyzing the decomposition of hydrogen peroxide (H_2_O_2_) into highly toxic hydroxyl radicals (·OH) using iron ions [86]. This therapy not only directly kills tumor cells but also activates the immune system by inducing immunogenic cell death (ICD), thereby promoting tumor-specific immune responses [87]. ICD, characterized by the release of tumor-specific antigens, activates dendritic cells and T cells, eliciting an immune response against the tumor. Moreover, the co-delivery of specific drugs, such as dihydroartemisinin (DHA) and pyropheophorbide-iron (Pyro-Fe), can further enhance the efficacy of ferroptosis. This co-delivery strategy not only increases the concentration of drugs at the tumor site but also enhances ferroptosis induction by generating more ROS [88]. This enhanced ferroptotic response can more effectively activate T cells, especially CD8+ T cells, which are crucial for antitumor immunity [89]. Figure 2 illustrates the interconnected components involved in enhancing CRC immunotherapy through ferroptosis modulation.

The expression patterns of ferroptosis-related genes provide new prognostic tools for the treatment of CRC, holding significant implications for personalized medicine and precision therapy. Studies investigating OFRG expression patterns have revealed different molecular clusters associated with clinical outcomes and biological pathways [78]. These molecular clusters help distinguish between “cold” and “hot” tumors in CRC. “Cold” tumors, characterized by low immune cell infiltration and poor response to immunotherapy, can be differentiated from “hot” tumors, which exhibit high immune cell infiltration and better response to immunotherapy [90]. This differentiation allows clinicians to select treatment regimens more accurately, providing patients with more effective treatments. Additionally, these studies suggest that the expression levels of ferroptosis-related genes can serve as biomarkers for predicting patient responses to PD-1-based immunotherapy [77,80]. For instance, high expression of ferroptosis-related genes may indicate a better response to PD-1 inhibitors, thereby facilitating more effective immunotherapy. The development of such predictive tools not only improves the success rate of treatment but also reduces unnecessary side effects and treatment costs. Some studies further emphasize the role of apolipoprotein L3 (*APOL3*) in ferroptosis and immunotherapy. APOL3 enhances the anti-tumor immune capacity of CD8+ T cells by promoting lactate dehydrogenase A (LDHA)-mediated ferroptosis [91]. This action increases the sensitivity of tumor cells to ferroptosis and promotes tumor immunogenic death by increasing interferon-gamma (IFNγ) production and reducing lactate concentration, enhancing tumor immunogenic death. This finding provides new predictive factors for the efficacy of immunotherapy and potential targets for the development of new treatment strategies.

Emerging treatment strategies incorporating the concept of ferroptosis into CRC immunotherapy offer new avenues for improving treatment efficacy and overcoming drug resistance [92]. The method discussed in studies involving the combination of ferroptosis inducers and immunotherapy demonstrates the potential of drug combinations and nanotechnology applications to enhance antitumor immune responses [93,94,95].

Of particular note is the strategy mentioned in a study, which involves the co-delivery of dihydroartemisinin (DHA) and pyropheophorbide-iron (Pyro-Fe). This co-delivery approach not only enhances the anti-cancer efficacy of DHA in vivo but also sensitizes non-immunogenic CRC tumors to PD-L1 checkpoint blockade immunotherapy by increasing tumor immunogenicity. The key to this strategy lies in the intervention of ferroptosis inducers, which alter the TME, promote the release and presentation of tumor antigens, activate dendritic cells and T cells, and trigger specific immune responses against the tumor [96]. Furthermore, the application of nanotechnology offers new possibilities for targeted delivery of ferroptosis inducers. Nanoparticles can be designed with specific targeting properties to deliver drugs more accurately to tumor cells while reducing damage to normal cells. This targeting not only enhances treatment efficacy but also reduces treatment side effects. For example, ZnP@DHA/Pyro-Fe core-shell nanoparticles mentioned in the literature can be prepared using nanotechnology to release DHA and Pyro-Fe at the tumor site, effectively inducing ferroptosis [97].

Recent advancements in understanding the molecular mechanisms of ferroptosis in CRC have shed light on its potential as a therapeutic target. The cellular processes of iron metabolism and lipid peroxidation are central to ferroptosis induction, with agents like elastin and sulfasalazine effectively triggering this form of cell death in CRC cells. This occurs through the inhibition of the xCT system, leading to significant changes within the cell, including reduced cystine uptake, glutathione depletion, and increased lipid peroxidation, offering a novel approach to cancer treatment [98,99].

One of the most promising aspects of ferroptosis research is its potential to overcome drug resistance in CRC. By specifically targeting ferroptosis regulators such as *GPX4* and *ACSL4*, alongside pathways involved in iron metabolism, researchers have been able to increase the sensitivity of CRC cells to both chemotherapy and immunotherapy [100]. This approach not only enhances the efficacy of these treatments but also reverses established drug resistance, marking a significant advancement in CRC treatment strategies. Furthermore, the interplay between ferroptosis and immunotherapy has opened new avenues for enhancing antitumor immunity [101]. The process of ferroptosis in tumor cells leads to the release of DAMPs, which can activate the immune system and promote antitumor responses. Particularly, the induction of ferroptosis has been shown to bolster the activity of CD8+ T cells within the TME, thereby amplifying the effectiveness of immunotherapeutic approaches [79].

Ferroptosis is an emerging form of cell death that has garnered significant attention for its potential in cancer treatment. Here is a summary of clinical studies involving ferroptosis-related molecules and drugs in CRC (https://clinicaltrials.gov). Sulfasalazine targets SLC7A11 and is being investigated in patients with metastatic CRC under clinical trial ID NCT06134388. The study aims to explore the effectiveness of sulfasalazine in this patient group. CNSI-Fe (II), which targets iron ions, is the focus of clinical trial NCT06048367. This study aims to evaluate the efficacy of carbon nanoparticle-loaded iron (CNSI-Fe (II)) in treating advanced solid tumors, including CRC. Neratinib, another drug targeting iron ions, is being examined in combination with Trastuzumab or Cetuximab for patients with KRAS/NRAS/BRAF/PIK3CA wild-type metastatic CRC. This phase II trial, identified as NCT03457896, seeks to determine the effectiveness of these combinations. Lapatinib, which also targets iron ions, is being evaluated in several studies for advanced or metastatic CRC. Clinical trials such as NCT00536809 (Phase I/II) [102], NCT00574171 (Phase II), NCT01184482 (Phase II), NCT04831528 (Phase II), and NCT03418558 (Phase II) are investigating Lapatinib in combination with various drugs. Some studies suggest that these combinations show efficacy in specific cancer types, but further research is needed to confirm their effectiveness in CRC.

Sorafenib, targeting *SLC7A11*, is involved in multiple clinical trials, including NCT00780169 (Phase I), NCT00869570 (Phase I) [103], NCT00989469 (Phase II) [104], NCT01715441 (Phase II) [105], NCT00826540 (Phase II) [106], NCT01290926 (Phase II) [107], NCT00134069 (Phase I/II), NCT00865709 (Phase II), NCT00703638 (Phase I), and NCT01376453 (Phase II) [108]. These studies investigate the efficacy of Sorafenib combined with different chemotherapy regimens for metastatic or locally advanced CRC. Some trials indicate that Sorafenib combination therapies are active and have acceptable toxicity, particularly in KRAS-mutant CRC. Simvastatin, targeting HMGCR, is the subject of clinical trial NCT01238094. This study assesses the efficacy of Simvastatin combined with XELIRI/FOLFIRI regimens in patients with metastatic CRC. Detailed information about these clinical trials can be found in Table 2.

Overall, Sulfasalazine and Sorafenib target SLC7A11, inhibiting glutamine transport and reducing the synthesis of the antioxidant *GSH*, thereby enhancing cellular sensitivity to oxidative stress and promoting ferroptosis [109,110]. CNSI-Fe (II) increases intracellular iron ion concentration, promoting ROS generation and lipid peroxidation, directly inducing ferroptosis [111]. Neratinib and Lapatinib studied primarily in combination with other therapies, show some preliminary efficacy in metastatic CRC, but further research is necessary to confirm their mechanisms and effectiveness. Simvastatin, though targeting HMGCR, might have potential efficacy when combined with chemotherapy in CRC. Research on ferroptosis-related therapeutic strategies in CRC is still in its early stages. Initial findings are promising, but extensive clinical studies are needed to validate these treatments’ efficacy and safety further [98].

## 7. Global Research Trends

From 2019 to 2024, there has been a significant increase in the number of publications on ferroptosis, particularly in the context of cancer research. The number of citations has also risen, indicating the high impact and relevance of these studies. Notable contributions have been made by researchers from the United States, China, and European countries, reflecting a broad international interest [112]. A substantial portion of the research focuses on understanding the biochemical pathways and molecular mechanisms underlying ferroptosis. This includes the roles of lipid peroxidation, iron metabolism, and antioxidant defenses in regulating cell death [20,49]. Another major theme is the exploration of ferroptosis as a therapeutic strategy. Studies investigate the potential of ferroptosis inducers to enhance the efficacy of existing cancer treatments, including chemotherapy, radiotherapy, and immunotherapy [113].

Research has also concentrated on identifying biomarkers for ferroptosis sensitivity and resistance. These biomarkers can help predict patient responses to ferroptosis-based therapies and tailor personalized treatment plans [101]. There is ongoing work in the development of novel ferroptosis inducers and inhibitors. This includes both synthetic compounds and natural products, with a focus on improving specificity and reducing side effects [114]. The field has seen robust international collaboration, with multi-institutional studies becoming more common. Collaborative networks often include partnerships between academic institutions, research hospitals, and pharmaceutical companies. This multidisciplinary approach is essential for translating basic research findings into clinical applications [115].

The future of research on ferroptosis in CRC immunotherapy is poised to address several critical areas. Further studies are needed to optimize the combination of ferroptosis inducers with other therapeutic modalities, such as ICIs. Translational research efforts are expected to focus on conducting clinical trials to evaluate the safety and efficacy of ferroptosis-based therapies in CRC patients. The development of advanced diagnostic tools to monitor ferroptosis in real time and assess treatment responses is anticipated. In-depth studies to understand and overcome resistance mechanisms to ferroptosis will be crucial for enhancing therapeutic outcomes.

By analyzing global research trends, it is evident that the study of ferroptosis in CRC immunotherapy is a rapidly evolving field with significant potential for improving cancer treatment. Continued international collaboration and multidisciplinary research will be key drivers of innovation and clinical translation.

## 8. Challenges and Limitations

One of the primary challenges in leveraging ferroptosis for CRC immunotherapy is the specificity and targeting of ferroptosis inducers. While ferroptosis has been shown to effectively induce cell death in tumor cells, ensuring that these inducers selectively target cancer cells without affecting normal cells is crucial. The off-target effects can lead to unwanted toxicity and damage to healthy tissues, which limits the clinical application of ferroptosis inducers [100]. The TME plays a significant role in modulating ferroptosis. Hypoxia, a common feature of the TME, can inhibit ferroptosis through various mechanisms, including the upregulation of hypoxia-inducible factors and iron metabolism regulators like IRP2 and CAIX. These factors protect cancer cells from ferroptosis, making it difficult to induce ferroptosis effectively in a hypoxic environment [113,116]. Cancer cells can develop resistance to ferroptosis through several pathways. For instance, the upregulation of antioxidant systems like the cystine/glutamate antiporter (system Xc-) and *GPX4* can counteract lipid peroxidation and prevent ferroptotic cell death [117]. Additionally, genetic alterations in ferroptosis-related genes and compensatory metabolic pathways can also contribute to resistance, posing a challenge to sustained therapeutic efficacy [118].

While ferroptosis can influence the immune response, it does not always result in an immunogenic form of cell death. Ferroptosis can inhibit dendritic cell maturation and antigen presentation, which are critical for initiating robust anti-tumor immune responses. Furthermore, the release of DAMPs during ferroptosis might not be sufficient to activate a strong anti-tumor immune response, limiting its effectiveness as an adjunct to immunotherapy [45]. Determining the optimal dosing and timing for ferroptosis inducers in combination with immunotherapy is another significant challenge. The therapeutic window must be carefully balanced to maximize tumor cell death while minimizing toxicity and adverse effects. Additionally, the timing of administration relative to immunotherapeutic agents needs to be optimized to achieve synergistic effects [56]. Translating preclinical findings into clinical practice requires robust biomarkers to identify patients who are likely to benefit from ferroptosis-based therapies. Currently, there is a lack of reliable biomarkers to predict ferroptosis sensitivity in CRC patients. The heterogeneity of tumors and the variability in ferroptosis-related gene expression further complicate patient stratification and the design of personalized treatment regimens [119]. Combining ferroptosis inducers with existing immunotherapies (e.g., checkpoint inhibitors) holds promise but also presents challenges. The interactions between different therapeutic modalities need to be thoroughly understood to avoid antagonistic effects. Additionally, the development of effective co-delivery systems for ferroptosis inducers and immunotherapeutic agents is crucial to enhance treatment efficacy and reduce systemic toxicity [120]. Finally, the development and approval of new ferroptosis-based therapies face significant regulatory hurdles. Ensuring the safety, efficacy, and quality of these therapies through rigorous clinical trials is essential. Long-term studies are required to assess potential late-onset toxicities and the overall impact on patient survival and quality of life [121]. Addressing these challenges requires a multidisciplinary approach, integrating insights from molecular biology, immunology, pharmacology, and clinical sciences. Continued research and innovation in this field are essential to overcome these limitations and harness the full potential of ferroptosis in CRC immunotherapy.

## 9. Future Prospect

The clinical application prospects of ferroptosis-related molecules as potential therapeutic targets represent a burgeoning area of research, particularly in the treatment of CRC [122]. Ferroptosis, a form of programmed cell death induced by lipid peroxidation, is associated with various pathophysiological processes and disease states, including cancer. In tumor therapy, modulation of key molecules related to ferroptosis can enhance the sensitivity of tumor cells to ferroptosis, thereby serving as a novel therapeutic strategy to improve the efficacy of immunotherapy.

Several key ferroptosis-related molecules and their potential clinical applications include *SLC3A2* and *SLC7A11*, which encode proteins comprising the system Xc- involved in the exchange of intracellular cystine and glutamate. Cystine is a crucial antioxidant that maintains the cellular reducing environment, while glutamate participates in cellular energy metabolism. Inhibiting *SLC3A2* and *SLC7A11* can decrease intracellular cystine levels in tumor cells, increasing their sensitivity to ferroptosis induction [29,36,123]. Therefore, drugs targeting these molecules may help induce ferroptosis in tumor cells and enhance the efficacy of immunotherapy.

Another significant molecule is *GPX4*, an antioxidant enzyme essential for maintaining cell membrane integrity and preventing lipid peroxidation. Overexpression of *GPX4* can protect tumor cells from the effects of ferroptosis. Therefore, inhibiting *GPX4* activity or expression may increase the sensitivity of tumor cells to ferroptosis, making them more susceptible to clearance by the immune system [124].

Ferroptosis inducers such as elastin and sorafenib can directly induce ferroptosis in tumor cells by interfering with intracellular iron metabolism and lipid peroxidation pathways, leading to tumor cell death [125]. In clinical treatment, these ferroptosis inducers can be used in combination with ICIs to enhance anti-tumor immune responses. Combination therapy strategies involving ferroptosis inducers and other treatment modalities such as radiotherapy, chemotherapy, or immunotherapy can enhance treatment efficacy. For example, certain chemotherapy drugs can increase tumor cell sensitivity to ferroptosis while inducing tumor cell death, thereby enhancing subsequent immunotherapy efficacy [126]. Furthermore, the development of biomarkers based on ferroptosis-related molecules can predict tumor responses to treatment. For instance, the expression levels of *SLC3A2* and *SLC7A11* may correlate with tumor sensitivity to ferroptosis inducers, serving as predictive factors for treatment outcomes.

In conclusion, the clinical application prospects of ferroptosis-related molecules lie in the development of novel treatment strategies. By modulating the expression or activity of these molecules, tumor cell sensitivity to ferroptosis can be enhanced, thereby improving the efficacy of immunotherapy. Future research needs to further explore the roles of these molecules in different types of tumors and evaluate their safety and effectiveness as therapeutic targets.

## 10. Summary

In conclusion, the relationship between ferroptosis and the immune system presents a promising frontier in colorectal cancer (CRC) therapy. By integrating ferroptosis inducers with current immunotherapies, we can enhance anti-tumor efficacy and overcome resistance. Key future directions include developing ferroptosis-based diagnostic tools to detect early tumor changes in the microenvironment, using ferroptosis inducers alongside immunotherapies to target resistant cancer cells, and identifying reliable biomarkers to predict ferroptosis sensitivity, optimizing treatment windows, and understanding therapeutic interactions. Conducting extensive research and clinical trials is crucial to translating these strategies into effective treatments.

Ferroptosis not only kills tumor cells but also enhances the immune system’s ability to recognize and attack them. By understanding the expression profiles of ferroptosis-related genes, we can tailor treatments more precisely, distinguishing between ‘cold’ and ‘hot’ tumors to optimize therapy. This approach aligns with the principles of precision medicine, allowing clinicians to navigate treatment plans with greater accuracy and efficacy.

However, challenges remain, including understanding the molecular interactions between ferroptosis and immune responses and integrating these insights into clinical practice. The journey from bench to bedside requires a nuanced understanding of the underlying biological mechanisms and careful consideration of the therapeutic window to balance efficacy with safety. Continued research and collaboration among the scientific and medical communities are essential to fully realize the potential of ferroptosis in CRC therapy. By leveraging the tumor’s own vulnerabilities and enhancing the immune response, we edge closer to a new era in cancer treatment where the demise of tumor cells is orchestrated by a synergistic approach combining ferroptosis and immunotherapy.

## Figures and Tables

**Figure 1 ijms-25-09141-f001:**
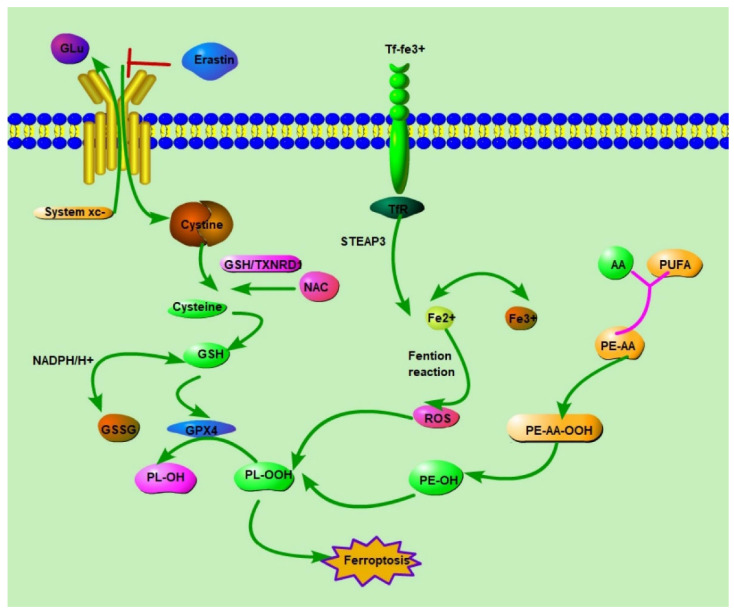
This figure shows key pathways and molecules involved in ferroptosis in tumor cells. System xc- imports cystine in exchange for glutamate, maintaining glutathione (*GSH*) levels, which act as antioxidants. *GPX4* uses *GSH* to prevent lipid peroxidation. Erastin inhibits system xc-, reducing cystine uptake and *GSH* and increasing lipid peroxidation. The transferrin receptor (TfR) uptakes iron, which, through the Fenton reaction, produces ROS that promotes lipid peroxidation. Polyunsaturated fatty acids (*PUFAs*) and arachidonic acid (AA) contribute to ferroptosis through lipid peroxidation. Antioxidants like N-acetylcysteine (*NAC*) and *GSH/TXNRD* reduce oxidative stress, inhibiting ferroptosis. This balance between lipid peroxidation and antioxidant defenses determines cell fate through ferroptosis. Abbreviations: *GSH*: Glutathione; *GPX4*: Glutathione peroxidase 4; GSR: Glutathione reductase; TXNRD1: Thioredoxin reductase 1; NAC: N-Acetylcysteine; *CoQ10*: Coenzyme Q10; FSP1: Ferroptosis suppressor protein 1; GCH1: GTP cyclohydrolase 1; BH4: Tetrahydrobiopterin; PUFA: Polyunsaturated fatty acid; PE-PUFA-OOH: Phosphatidylethanolamine-linked PUFA hydroperoxides; α-TOH: Alpha-tocopherol (Vitamin E); IPP: Isopentenyl pyrophosphate; CoA: Coenzyme A; BCNU: Carmustine; RSL3: RAS-selective lethal 3.

**Figure 2 ijms-25-09141-f002:**
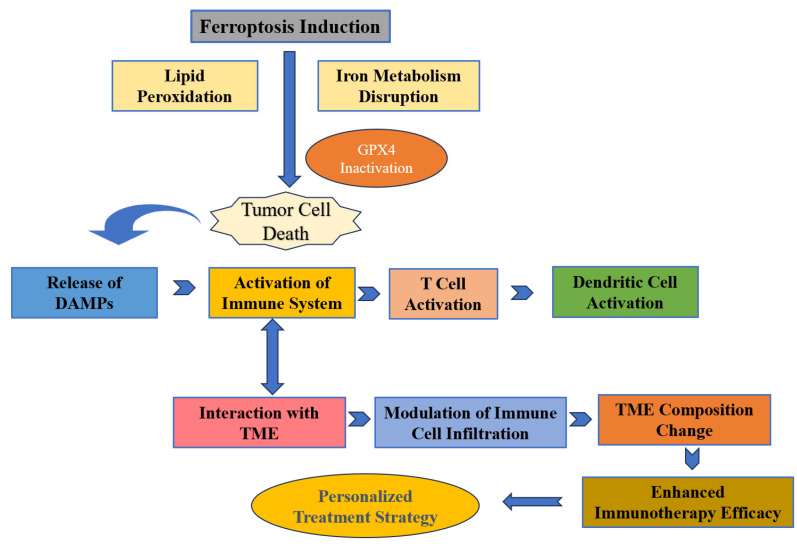
The figure depicts the sequence of events from ferroptosis induction to its impact on tumor microenvironment (TME) and immunotherapy. Ferroptosis triggers lipid peroxidation and *GPX4* inactivation, leading to tumor cell death and DAMPs release. This activates the immune system, enhancing T cell and dendritic cell functions, and modulates TME for improved immunotherapy efficacy, paving the way for personalized treatment strategies.

**Table 1 ijms-25-09141-t001:** Regulatory Interplay between Ferroptosis and the Tumor Microenvironment: Mechanisms and Impacts.

TME Factor/ Regulator	Mechanism of Impact	Effect on Ferroptosis	Reference Numbers
Hypoxia	Induction of IRP2 post-translational upregulation<br>- Regulation of iron transporter and storage protein expression	Promotes ferroptosis	[26]
Carbonic Anhydrase IX (CAIX)	Modulation of the cystine/glutamate antiporter xCT	Prevents ferroptosis	[27,28]
HIF-1α/lncRNA-PMAN	Regulation of key genes involved in iron metabolism	Protects cancer cells from ferroptosis	[29]
CBSLR/CBS Signal Axis	Same as above	Protects cancer cells from ferroptosis	[29]
NCOA4 Expression Reduction	Reduced expression in macrophages, leading to increased ferritin levels and decreased ferroptosis susceptibility	Decreases ferroptosis susceptibility	[22,30]
Polymeric Nanocarriers	Depleting *NADPH*, *GSH*, and Trx to sensitize hypoxic tumor cells to ferroptosis	Sensitizes hypoxic tumor cells to ferroptosis	[31,32]
Lactate Accumulation	Upregulation of HCAR1 and MCT1 reducing lipid peroxidation	Inhibits ferroptosis	[33]
Inflammatory Factors	Regulation of ferroptosis through various cancer types and signaling pathways, such as *JAK2/STAT3* and *RAS-JNK/p38*	May promote or inhibit ferroptosis	[34]
Non-coding RNAs	Modulation of ferroptosis by targeting key regulators like xCT and *GPX4*	Influences cancer progression and ferroptosis	[35]
CD8 T Cells	Secretion of cytokines like *IFNγ* inhibiting the expression of *SLC3A2* and *SLC7A11*	Promotes ferroptosis in tumor cells	[36,37]
oxLDL and CD36	Inducing ferroptosis and p38 phosphorylation in CD8+ T cells via CD36-dependent mechanisms	Affects T cell function and antitumor immunity	[40,41,42]
NK Cells	Cytotoxic effects on tumor cells through perforin, granzyme, and *IFNγ* release	Enhances NK cell activity and promotes ferroptosis	[42,43]
Ferroptosis and Immune Response	Release of cytokines and DAMPs without activating antitumor immune responses, inhibiting dendritic cell functions	May not represent an immunogenic cell death form	[44,45]

**Table 2 ijms-25-09141-t002:** Clinical Studies on Ferroptosis-related Molecules and Drugs in CRC.

Drug/Molecule	Target	Clinical Trial ID	Phase	Combination/Specific Aim	Reference
Sulfasalazine	*SLC7A11*	NCT06134388	Phase I	Metastatic CRC	-
CNSI-Fe (II)	Iron ions	NCT06048367	Phase I	Advanced solid tumors including CRC	-
Neratinib	Iron ions	NCT03457896	Phase II	Combination with Trastuzumab or Cetuximab	-
Lapatinib	Iron ions	NCT00536809 NCT00574171 NCT01184482 NCT04831528 NCT03418558	Phase I/II	Combination with various drugs	[102]
Sorafenib	*SLC7A11*	NCT00780169 NCT00869570 NCT00989469 NCT01715441 NCT00826540 NCT01290926 NCT00134069 NCT00865709 NCT00703638 NCT01376453	Phase I/II	Combination with chemotherapy	[103,104,105,106,107,108]
Simvastatin	HMGCR	NCT01238094	Phase I	Combination with XELIRI/FOLFIRI	-

Note: “-” indicates that the clinical research is still ongoing, and there are no related publications available at this time.

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
