# Peer review of "Enhancing Colorectal Cancer Immunotherapy: The Pivotal Role of Ferroptosis in Modulating the Tumor Microenvironment"

_ijms, 2024, doi:10.3390/ijms25179141_

Round 1

Reviewer 1 Report

Comments and Suggestions for Authors

In this study, Li and Cheng explain how the relationship between ferroptosis and the immune system represents a promising frontier in colorectal cancer immunotherapy. The authors detail how ferroptosis can not only induce cancer cell death, but also reshape the tumour microenvironment to enhance the efficacy of immunotherapy.

The manuscript is clear and well written and includes all data from the recent literature. The figures help the reader to understand the different cellular mechanisms considered. I wish the authors good luck in the publication process.

Minor revisions: 

It would be very useful to include a section in this review that takes into account the sex and gender differences that exist in the development of CRC and the response to immunotherapy. This would not only make the review more interesting, but also give the reader a more complete overview of the subject.

Lane 15: OFRGs is reported as an abbreviation. Please add the full name as it appears to the line 27

Lane 185: DAMPs is reported as an abbreviation only. Please add the full name as it appears to the line 256

Figure 2: There is no correspondence between the arrows in the figure and the description in the caption. Authors are advised to use a different illustration, which may help the reader to interpret the various links between ferroptosis, immunotherapy and the tumour microenvironment. 

Comments on the Quality of English Language

I suggest to the authors a minimal revision of English

Author Response

Manuscript ID: ijms-3156058

Title: Enhancing Colorectal Cancer Immunotherapy: The Pivotal Role of Ferroptosis in Modulating the Tumor Microenvironment

Dear Editor,

We thank you and the anonymous reviews very much for the constructive criticism. Please find below the amendments that we have made in this revision.

I will be glad to receive further editorial correspondence through this email: [email protected]

Best regards,

Dr. Xiaofei Cheng

Responds to the reviewer’s comments:

Thank you for the opportunity to revise and improve our manuscript. We greatly appreciate the positive feedback and the recognition of our work's clarity and relevance. Your comments encourage us as we continue to explore the important relationship between ferroptosis and the immune system in the context of colorectal cancer immunotherapy. We are pleased to hear that the manuscript is well-written and that the figures effectively support the understanding of the cellular mechanisms discussed. We are confident that the revisions based on your insightful suggestions have strengthened our work, and we look forward to the potential publication of our study.

  1. Response to comment: It would be very useful to include a section in this review that takes into account the sex and gender differences that exist in the development of CRC and the response to immunotherapy. This would not only make the review more interesting, but also give the reader a more complete overview of the subject.

Response: We appreciate the reviewer’s insightful suggestion to include a discussion on sex and gender differences in the development of CRC and the response to immunotherapy. We agree that this is an important aspect that could provide a more comprehensive understanding of the topic. In response, we have added a new section to the manuscript that addresses the influence of sex and gender on CRC progression and how these factors may affect the efficacy of immunotherapy. We believe this addition enriches the review by offering a broader perspective on the subject.

Sex differences in the progression of CRC have become an increasingly important area of research. Studies show that men and women exhibit significant differences in CRC incidence, progression, and response to treatment. Generally, men have a higher incidence of CRC, which may be linked to lifestyle-related risk factors such as smoking, alcohol consumption, and diet. In contrast, women may benefit from protective effects of hormones like estrogen, which may delay CRC onset. However, this protective effect diminishes after menopause, leading to a convergence in CRC risk between men and women.  At the genetic level, there are sex-based differences in the frequency and types of mu-tations in CRC-related genes, such as APC and TP53, which can influence tumor be-havior and treatment response. Estrogen, for example, is thought to exert anti-tumor effects by interacting with intracellular signaling pathways that inhibit CRC develop-ment. Additionally, estrogen may modulate the immune microenvironment, poten-tially enhancing the effectiveness of immunotherapy. Sex differences also extend to the immune response, with women generally exhibiting a stronger immune reaction compared to men. This heightened immune response could lead to better outcomes in immunotherapy for women but also increases the risk of immune-related side effects.

  1. Response to comment: Lane 15: OFRGs is reported as an abbreviation. Please add the full name as it appears to the line 27.

Response:  Thank you for pointing this out. We have revised the manuscript to include the full name of the abbreviation OFRGs on line 15. The full name is now provided as " Ferroptosis-related Genes ".

  1. Response to comment: Lane 185: DAMPs is reported as an abbreviation only. Please add the full name as it appears to the line 256.

Response: Thank you for your observation. We have updated the manuscript to include the full name of the abbreviation DAMPs on line 204. The full name, "Damage-Associated Molecular Patterns," is now provided to ensure clarity for the readers. We appreciate your assistance in improving the manuscript.

  1. Response to comment: Figure 2: There is no correspondence between the arrows in the figure and the description in the caption. Authors are advised to use a different illustration, which may help the reader to interpret the various links between ferroptosis, immunotherapy and the tumor microenvironment.

Response: We appreciate your feedback regarding Figure 2. We have revised the figure to better align with the description in the caption and to improve the clarity of the connections between ferroptosis, immunotherapy, and the tumor microenvironment. A new illustration has been created to more accurately represent these relationships, making it easier for readers to interpret the various links. The updated figure and caption are now included in the revised manuscript.

Reviewer 2 Report

Comments and Suggestions for Authors

In the manuscript, the authors concentratedly reviewed the recent development on enhancing colorectal cancer immunotherapy with ferroptosis. A quite a lot examples have been selected, explained and discussed comprehensively by the authors in order to provide readers a better understanding on the area of ferroptosis-immunotherapy, in which new therapeutic strategies may be developed in the future that could enhance further efficacy and broader applicability in CRC treatment. In general, the manuscript is well-organized in a logical and systematic way. The references selected are up-to-date. This study may be attractive to a broad spectrum of readers in the field of anticancer.

There are some minor issues found and that the authors may need to double check for its correctness.

-          Introduction: “Colorectal cancer (CRC) is among of the cancers with the highest incidence and mortality rates worldwide”. These data stated by the authors may not be correct. The authors are strongly recommended to check from WHO for detailed and up-to-date information.

-          For the figures used, it is unclear whether permission is required.

-          Figure 2: More details could be given in each items, for example TME (x,y,z,…)

-          Table 2: It is unclear why some without reference(s) given.

-          Could the authors also list some important checkpoint inhibitors? Particularly, those show therapeutic potential in combining with ferroptosis inducers.

Author Response

Manuscript ID: ijms-3156058

Title: Enhancing Colorectal Cancer Immunotherapy: The Pivotal Role of Ferroptosis in Modulating the Tumor Microenvironment

Dear Editor,

We thank you and the anonymous reviews very much for the constructive criticism. Please find below the amendments that we have made in this revision.

I will be glad to receive further editorial correspondence through this email: [email protected]

Best regards,

Dr. Xiaofei Cheng

Responds to the reviewer’s comments:

We sincerely thank you for your positive feedback and for recognizing the comprehensive nature of our review. We are pleased to hear that you found the manuscript well-organized and the references up-to-date. Your acknowledgment of the potential impact of our work in advancing therapeutic strategies for colorectal cancer is greatly appreciated. We value your input and are encouraged that our study may contribute to the broader field of anticancer research.

  1. Response to comment: Introduction: Colorectal cancer (CRC) is among of the cancers with the highest incidence and mortality rates worldwide. These data stated by the authors may not be correct. The authors are strongly recommended to check from WHO for detailed and up-to-date information.

Response: Thank you for pointing out this issue. We have reviewed the latest data from the World Health Organization (WHO) and revised the introduction accordingly. We appreciate your attention to ensuring the accuracy of these important details.

  1. Response to comment: For the figures used, it is unclear whether permission is required.

Response:  Thank you for highlighting this concern. We would like to clarify that Figure 1 was created using Pathway Builder software, and we have verified that no specific permissions are required for its use. Figure 2 has been newly created and redesigned using PPT. We have ensured that all figures comply with the necessary permissions and guidelines.

  1. Response to comment: Figure 2: More details could be given in each items, for example TME (x,y,z,).

Response: Thank you for your valuable suggestion. We have revised Figure 2 to include more detailed information for each item. Additionally, we have updated the figure legend to provide a more comprehensive description of the figure's content. We believe these changes enhance the clarity and detail of the figure, improving its overall usefulness.

  1. Response to comment: Table 2: It is unclear why some without reference(s) given

Response: Thank you for highlighting this issue. The lack of references for some entries in Table 2 is due to the fact that the clinical studies mentioned are still ongoing and have not yet been published. We have added a note to the table to clarify this situation. We appreciate your understanding and attention to this matter.

  1. Response to comment: Could the authors also list some important checkpoint inhibitors? Particularly, those show therapeutic potential in combining with ferroptosis inducers.

Response: Thank you for your suggestion. We have addressed this topic in the section titled "The Role of Ferroptosis in Immunotherapy for CRC." In the context of colorectal cancer, important checkpoint inhibitors such as anti-PD-1 antibodies (Pembrolizumab and Nivolumab), anti-PD-L1 antibodies (Atezolizumab, Durvalumab, and Avelumab), and anti-CTLA-4 antibody (Ipilimumab) have been extensively studied and used in clinical treatments. The combination of ferroptosis inducers with these checkpoint inhibitors may exhibit significant synergistic effects. For instance, the combination of anti-PD-1 antibodies like Pembrolizumab with ferroptosis inducers (such as RSL3 or Ferrostatin-1) has demonstrated enhanced anti-tumor effects in colorectal cancer models.
